# Adults vs. neonates: Differentiation of functional connectivity between the basolateral amygdala and occipitotemporal cortex

**Heather A. Hansen**[ID]*, **Jin Li, Zeynep M. Saygin***

Department of Psychology, The Ohio State University, Columbus, Ohio, United States of America

* hansen.508@osu.edu (HAH); saygin.3@osu.edu (ZMS)

**Data Availability Statement:** The data used in this study are publicly available. All relevant accession codes are publicly available for both HCP (https://www.humanconnectome.org/study/hcp-young-

## Abstract

The amygdala, a subcortical structure known for social and emotional processing, consists of multiple subnuclei with unique functions and connectivity patterns. Tracer studies in adult macaques have shown that the basolateral subnuclei differentially connect to parts of visual cortex, with stronger connections to anterior regions and weaker connections to posterior regions; infant macaques show robust connectivity even with posterior visual regions. Do these developmental differences also exist in the human amygdala, and are there specific functional regions that undergo the most pronounced developmental changes in their connections with the amygdala? To address these questions, we explored the functional connectivity (from resting-state fMRI data) of the basolateral amygdala to occipitotemporal cortex in human neonates scanned within one week of life and compared the connectivity patterns to those observed in young adults. Specifically, we calculated amygdala connectivity to anterior-posterior gradients of the anatomically-defined occipitotemporal cortex, and also to putative occipitotemporal functional parcels, including primary and high-level visual and auditory cortices (V1, A1, face, scene, object, body, high-level auditory regions). Results showed a decreasing gradient of functional connectivity to the occipitotemporal cortex in adults–similar to the gradient seen in macaque tracer studies–but no such gradient was observed in neonates. Further, adults had stronger connections to high-level functional regions associated with face, body, and object processing, and weaker connections to primary sensory regions (i.e., A1, V1), whereas neonates showed the same amount of connectivity to primary and high-level sensory regions. Overall, these results show that functional connectivity between the amygdala and occipitotemporal cortex is not yet differentiated in neonates, suggesting a role of maturation and experience in shaping these connections later in life.

## Introduction

How does emotional valence influence visual perception? Whether it be driving by an emotionally salient car crash or happening upon an animal carcass in the jungle, perceiving visual

adult) and dHCP (http://www.
developingconnectome.org/)

**Funding:** Financial support was provided by the
Alfred P. Sloan Foundation (to Z.M.S) and Ohio
State's Chronic Brain Injury Program (to Z.M.S).
The funders had no role in study design, data
collection and analysis, decision to publish, or
preparation of the manuscript.

**Competing interests:** The authors have declared
that no competing interests exist.

stimuli through an emotional lens can be critical for quick motor responses and ultimate survival. Emotionally salient cues preceding a target can enhance target perception (e.g., [1]), and perceiving aversive stimuli enhances blood flow to cortical regions (e.g., the middle temporal gyrus [2]). Developmentally, not only does visual acuity improve with age [3], but visual perceptual mechanisms of emotional stimuli are also fine-tuned with experience (e.g., [4]).

Emotional valence is canonically tied to the amygdala, an evolutionarily preserved neural structure known for emotional processing and regulation (e.g., [5, 6]). The amygdala has been additionally implicated in social cognition and attention (e.g., [7]), fear recognition and conditioning (e.g., [8, 9]), stimulus-value learning and reward (e.g., [10, 11]), and novelty detection (e.g., [12, 13]). The functions of the amygdala and the way in which the amygdala assigns valence to stimuli change across development [14]. Similarly, visual perceptual skills and their neural correlates also change across development [15]. Perceiving the identity of visual stimuli is commonly attributed to the occipitotemporal cortex, the location of the ventral visual stream and "what" pathway (e.g., [16]). It is posited that emotionally enhanced visual perception may occur via cortical feedback connections between the amygdala and visual cortex [17].

Work in macaques shows that projections from the amygdala subnuclei to the ventral visual stream are topographically organized on a gradient, such that visual cortical areas that are more rostral receive heavier amygdalar projections than visual cortical areas that are more caudal [18, 19]. Amaral and colleagues [19–22] found the basal subnucleus of the amygdala to especially follow this pattern, but noted additional projections from area TE to the lateral subnucleus that creates a feedforward/feedback loop. Other work in adult macaques has similarly shown projections from areas TEO and TE to the lateral nucleus of the amygdala, and from area TE to the basal nucleus [23].

Interestingly, these connections change over development. Experiments comparing adult to juvenile animals, specifically in nonhuman primates (e.g., [23–26]) and rats (e.g., [27, 28]), reveal that amygdalar projections are adult-like in juveniles, but that juveniles also have additional connections that are either totally eliminated with maturation or become more refined in their distribution.

Do these connections show a similar pattern in human development? We know that macaque cortex is oriented differently than human cortex, and although homologies exist, the connectivity pattern in macaques may not necessarily perfectly map to humans [29, 30]. Moreover, in humans, it is more challenging to study amygdalar connections at such a fine-grained level that tracer studies can provide, especially with respect to the basal vs. lateral nucleus and their connections to visual cortex. Several groups have used a variety of methods to parcellate the amygdala into two to four subunits (e.g., [31–37]). More recent work has made it possible to use local intensity differences in a typical T1 scan to divide the human amygdala into nine separate subunits [38], thus allowing a way to parcellate the amygdala using a standard resolution anatomical (T1) image and explore the connectivity of these subunits with a separate (independent) connectivity scan.

There is some previous work in humans that explores the developmental changes of amygdalar connectivity. A study that explored a cross-sectional sample of 5–30 year olds showed that DWI connectivity of the lateral and basal nuclei to cortical areas becomes increasingly sparse and localized with age [39]. A functional connectivity study in 7–9 year olds vs. adults found that the basolateral amygdala had stronger connectivity with temporal regions than the centromedial amygdala, and that overall connectivity was stronger in adults compared to children [34]. Another study showed that basolateral functional connectivity to regions including parahippocampal gyrus, superior temporal cortex, and occipital lobe decreases with age across 4-23-year-olds [36], but that the basolateral amygdala showed increasing functional connectivity to occipital cortex between ages 3 months to 5 years [37]. This last study is the opposite

pattern than what was found in macaque development (i.e., decreasing connectivity to occipital cortex across age, e.g., [23]), and may be due to the differences between functional vs. white-matter connectivity or due to differences between macaques and humans. Moreover, it remains unclear why these connections change with development; the occipitotemporal cortex contains a multitude of well-studied visual and auditory functional areas. It is possible that amygdala connectivity changes with respect to functionally specific parts of occipitotemporal cortex that show increasing developmental specialization. To date, no study has investigated neonatal functional connectivity of the amygdala subnuclei and no study has investigated this connectivity with respect to putative functionally-distinct regions within visual and auditory cortex.

Does the rostrocaudal gradient of connectivity from the basolateral subnucleus observed in macaques match that of humans, or will a different pattern emerge? Does this connectivity pattern exist from birth, or develop later in life? And are the developmental changes in connectivity specific to certain functional parcels located within the occipitotemporal cortex? Here we investigate the developmental changes in functional connectivity between the basolateral amygdala and the occipitotemporal cortex using a cross-sectional sample of adults and neonates. In the first set of analyses we target the entire occipitotemporal cortex to recreate the connectivity work done in macaques. Then, we apply a unique approach by targeting functionally defined regions in the ventral visual stream in order to draw conclusions about what might be driving the observed pattern of connectivity.

## Materials and methods

### Participants

**Neonates.**  Forty neonates (15 female, 25 male; mean gestational age at birth = 38.99 weeks, gestational age range at scan = 37–44 weeks) were obtained from the initial release of the Developing Human Connectome Project (dHCP, http://www.developingconnectome.org) [40]. Neonates were scanned at the Evelina Neonatal Imaging Center in London, and the study was approved by the UK Health Research Authority.

**Adults.**  Forty adults (15 female, 25 male; age range 22–36 years) were obtained from the Human Connectome Project (HCP), WU-Minn HCP 1200 Subjects Data Release (https://www.humanconnectome.org/study/hcp-young-adult) [41]. All participants were scanned at Washington University in St. Louis, MO. The forty adults used in this study were chosen to best motion- and sex-match the neonate sample: for each neonate, an adult from the HCP dataset with the same sex and most similar motion parameter (i.e., framewise displacement, FD) was determined using k-nearest neighbors. By using this approach, head motion in the final samples was not significantly different between groups ($t(78) = 0.77$, $p = 0.45$).

### Acquisition and preprocessing

**Neonates.**  Images were acquired on a Philips 3T Achieva scanner using a specially designed neonatal 32 channel phased array head coil with dedicated slim immobilization pieces to reduce gross motion [42]. All neonates (i.e., both MRI and fMRI scans) were scanned while in natural sleep. High-resolution ($0.8\ mm^3$) structural scans were acquired on all participants. T2-weighted and inversion recovery T1-weighted multi-slice fast spin-echo images were acquired with in-plane resolution $0.8 \times 0.8\ mm^2$ and 1.6 mm slices overlapped by 0.8 mm (T2-weighted: TE/TR = 156/12000ms; T1 weighted: TE/TR/TI = 8.7/4795/1740ms). Structural MRI data were preprocessed in FreeSurfer v.6.0.0 (http://surfer.nmr.mgh.harvard.edu/fswiki/infantFS) using a dedicated infant processing pipeline [40, 43, 44] which includes motion and intensity correction, surface coregistration, spatial smoothing, subcortical segmentation, and

cortical parcellation based on spherical template registration; FreeSurfer was used for amygdala segmentation [38]. Gray and white matter masks were obtained from segmentations of the T2w volume using the DRAW-EM algorithm provided by dHCP [45]. The resulting cortical and subcortical segmentations were reviewed for quality control.

Resting-state fMRI data were also acquired on all participants, using multiband (MB) 9x accelerated echo-planar imaging (TE/TR = 38/392ms, voxel size = 2.15 mm$^3$) developed for neonates (see [46] for details). The resting-state scan lasted approximately 15 min and consisted of 2300 volumes for each run. No in-plane acceleration or partial Fourier was used. Single-band reference scans were also acquired with bandwidth-matched readout, as well as additional spin-echo acquisitions with both AP/PA fold-over encoding directions. The data released by the dHCP included minimal preprocessing of the resting-state fMRI data (see [46]) which included distortion-correction, motion-correction, 2-stage registration of the MB-EPI functional image to the T2 structural image, generation of a combined transform from MB-EPI to the 40-week T2 template, temporal high-pass filtering (150 s high-pass cutoff), and independent component analysis (ICA) denoising using FSL FIX. Additional preprocessing included smoothing within gray matter (Gaussian filter with FWHM = 3 mm), and a band-pass filter at 0.009–0.08 Hz. To further denoise, aCompCor [47] was used to regress out signals from white matter and cerebrospinal fluid (CSF) which controls physiological noise (e.g., respiration, heartbeat) and non-neural contributions to the resting state signal. All functional connectivity analyses for the neonatal group were performed in native functional space.

**Adults.** Images were acquired on a customized 3T Connectome Scanner adapted from a Siemens Skyra (Siemens AG, Erlanger, Germany). The 32-channel scanner had a receiver head coil and a body transmission coil specifically designed by Siemens for the WU-Minn and MGH-UCLA Connectome scanners.

High-resolution T2-weighted and T1-weighted structural scans were acquired on all participants. Images were acquired with 0.7 mm$^3$ isotropic voxel resolution (T2-weighted 3D T2-SPACE scan: TE/TR = 565/3200ms; T1-weighted 3D MPRAGE: TE/TR/TI = 2.14/2400/1000ms). The data that were released had undergone preprocessing using the HCP minimal preprocessing pipelines (see [48] for details), which included: gradient distortion correction, ACPC registration to produce an undistorted "native" structural volume space, brain extraction, bias field correction, and registration from the T2-weighted scan to the T1-weighted scan. Each adult brain was aligned to a common MNI152 template with 0.7 mm isotropic resolution. Then, a FreeSurfer pipeline (based on FreeSurfer 5.3.0-HCP) specifically designed for HCP data was used to segment the volume into predefined structures, reconstruct white and pial cortical surfaces, and perform folding-based surface registration to their surface atlas (fsaverage).

Resting-state fMRI data were also acquired on all participants, using the gradient-echo EPI sequence (TE/TR = 33.1/720ms, flip angle = 52˚, number of slices = 72, voxel size = 2 × 2 × 2 mm$^3$). The resting-state scan lasted approximately 15 min and consisted of 1200 volumes for each run. All participants completed two resting-state fMRI sessions, each consisting of one run with two phases encoding in a right-to-left (RL) direction and one run with phase encoding in a left-to-right (LR) direction; the current analysis uses the LR phase encoding from the first session. Participants were instructed to open their eyes with relaxed fixation on a projected bright cross-hair on a dark background. The data that were released had undergone minimal preprocessing [48], which included removal of spatial distortions, motion correction, registration of the fMRI data to both the structural and MNI-152 template, bias field reduction, and denoising using the novel ICA-FIX method. In order to preprocess these data in a pipeline that mirrored the neonatal group, we unwarped the data from MNI-152 to native space, then applied spatial smoothing (Gaussian filter with FWHM = 3 mm) within all gray matter, band-pass filtered at 0.009–0.08 Hz, and implemented aCompCor [47].

### Defining regions of interest

**Amygdala subnuclei.** Using automated segmentation [38], nine amygdala subnuclei (lateral, basal, accessory basal, central, medial, cortical, paralaminar, cortico-amygdaloid transition area, anterior amygdala area) were parcellated in each individual's native anatomical space and then transferred to functional space. Because the lateral and basal subnuclei are associated with sensory and cognitive processes [49, 50] and thus likely contribute to emotional visual perception, the combined basolateral subnucleus was the main seed of interest in the present experiment. S1 Fig compares the basolateral amygdala segmentation in neonates with the dHCP-provided amygdala labels; we found that almost all of the basolateral amygdala used here was within the dHCP manually-labeled amygdala (proportion of BaLa within dHCP amygdala: 0.76 ± 0.11).

**Occipitotemporal cortex.** To explore the connectivity of the basolateral amygdala to occipitotemporal cortex (OTC), an OTC label was made for each individual using anatomical labeling provided by each data set (i.e., DRAW-EM labels for neonates, aparc+aseg labels for adults; see S1 File for labels used and S2 Fig for a depiction of labels in a neonate vs. adult) that combined all anatomical regions in the occipital and temporal cortices. The OTC label was transferred from native anatomical space to functional space for each subject. In order to track differences in connectivity across the region, the label was split (separately for each individual and each hemisphere) into five equal sections from anterior to posterior. These five anatomical OTC sections were the connectivity targets for the first analyses (see Fig 1A).

To explore the functional significance of connectivity patterns, functional parcels that encompass primary and secondary visual and auditory areas within the OTC were identified. All parcels that we used are available online and/or by contacting the corresponding author of the cited publications. The parcels were originally created via the group-constrained subject-specific method (GSS) [51], which generates probabilistic maps of functional activation across independent groups of participants and creates parcels that encapsulate most individuals' functional regions. We used the face-selective fusiform face area (FFA), occipital face area (OFA), and superior temporal sulcus (STS); object-selective lateral occipital cortex (LO) and posterior fusiform sulcus (PFS); scene-selective parahippocampal place area (PPA) and retrosplenial cortex (RSC) from [52]; and high-level auditory region superior temporal gyrus (STG), a region in vicinity of primary auditory cortex involved in speech perception [53]. In addition, primary visual cortex (V1) and auditory cortex (A1) were anatomically defined in each subject using the calcarine sulcus and Heschl's gyrus from FreeSurfer Desikan parcellation [54], respectively. See Fig 2A for an illustration of the parcels.

Functional parcels were mapped to the FreeSurfer CVS average-35 in MNI152 brain (if not already publicly provided in that space) and were subsequently overlaid onto each individual's anatomical brain using Advanced Normalization Tools (ANTs version 2.1.0; http://stnava.github.io/ANTs) [55]. The parcels were then converted to native functional space using nearest neighbor interpolation with FreeSurfer's mri_vol2vol function (https://surfer.nmr.mgh.harvard.edu/fswiki/mri_vol2vol). For any parcels that overlapped, intersecting voxels were assigned to the functional parcel with smaller size; this ensured that no voxel belonged to more than one functional parcel, and additionally compensated for size differences. Finally, voxels within white matter and cerebellum were removed.

### Functional connectivity analyses

The mean time course of the basolateral amygdala, each OTC section, and each functional parcel was computed from the preprocessed resting-state images. Functional connectivity (FC) was calculated using Pearson's correlations between the time courses of the basolateral seed

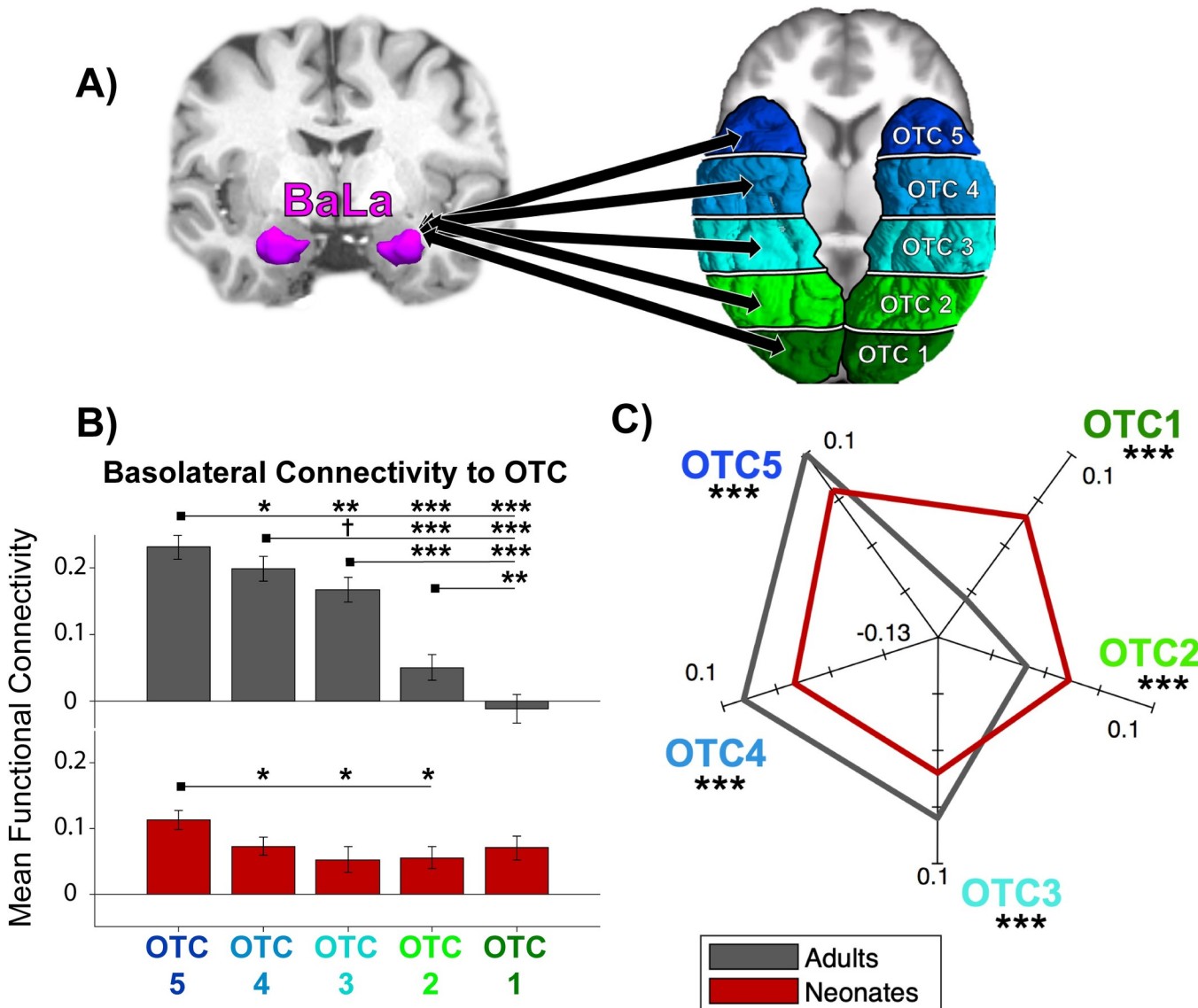

**Fig 1. Anatomical regions and functional connectivity results.** (A) Basolateral (BaLa) amygdala and anatomical targets used for connectivity analyses. Left, an example parcellation of the basal and lateral amygdala subnuclei in a representative subject, using the atlas developed by Saygin et al., 2017. Right, depiction of the 5 occipitotemporal cortex (OTC) labels in a representative subject. Labels marked from most anterior (OTC 5, dark blue) to most posterior (OTC 1, dark green). (B) Bar plot of mean functional connectivity to each of the 5 OTC sections arranged from anterior to posterior for each sample, with adults in gray and neonates in red. Error bars are standard error of the mean. $^{†}p<0.06$, $^{*}p<0.05$, $^{*}p<0.01$, $^{***}p<0.001$ (C) FC fingerprint plot depicting the pattern of connectivity of both samples. Axes are mean centered FC values for each sample.

and each target region, collapsed across hemispheres. To generate normally distributed values, each FC value was Fisher z-transformed.

Connectivity differences were calculated using 2-way mixed ANOVAs, with sample (adults vs. neonates) as the between-subject variable and target (i.e., different anatomical/functional regions of interest) as the within-subject variable. Paired *t*-tests were conducted for within-group comparisons and independent *t*-tests for between-group comparisons. The Holm-Bonferroni method was used to correct for multiple comparisons for each post-hoc test; corrected *p*-values are denoted as $p_{HB}$.

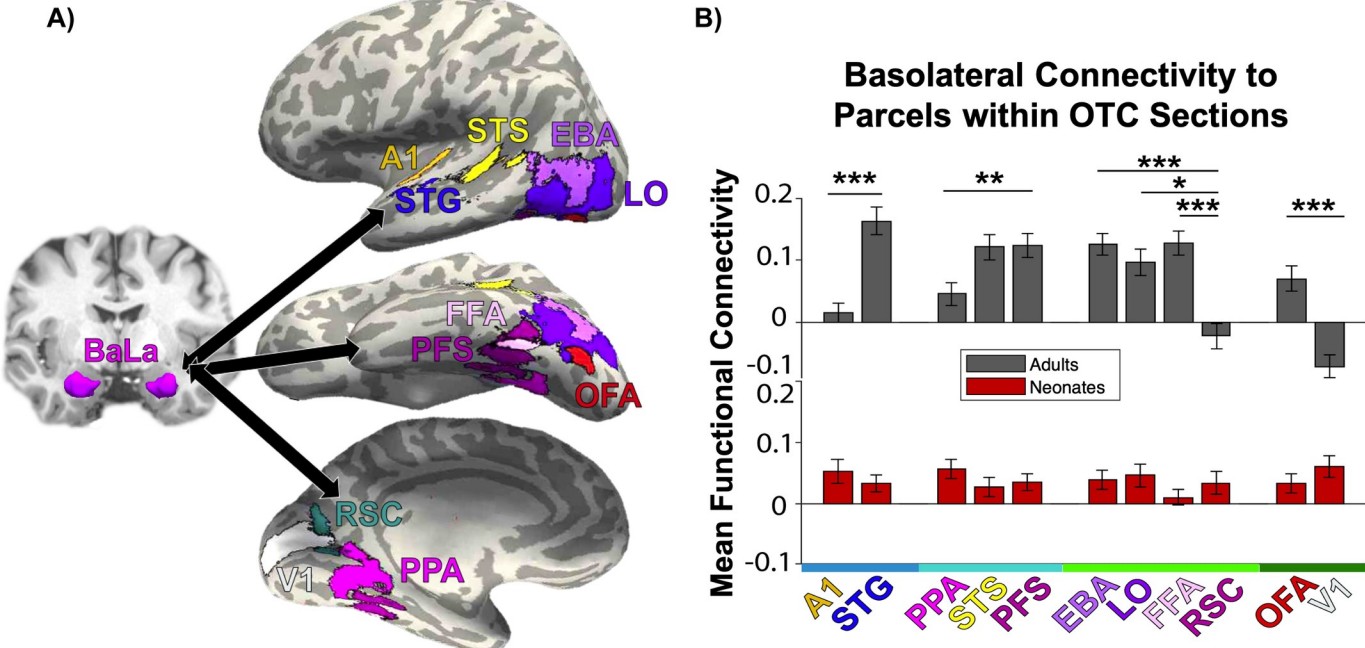

**Fig 2. Functional parcels and functional connectivity results.** (A) Basolateral (BaLa) amygdala and functional targets used for connectivity analyses. Left, BaLa parcellation in a representative subject. Right, depiction of the 11 functional parcels used as targets. (B) Bar plot of mean functional connectivity to each of the 11 parcels arranged from anterior to posterior for each sample, with adults in gray and neonates in red. X-axis color represents OTC section where majority of parcel is located, from blue (OTC 4, A1 and STG) to dark green (OTC 1, OFA and V1). Error bars are standard error of the mean. Significance depicted between regions within the same OTC section only. $^*p<0.05$, $^{**}p<0.01$, $^{***}p<0.001$.

Finally, we created FC fingerprint plots to elucidate between-group differences. For each set of targets, connectivity values were mean-centered across subjects in each sample by subtracting the mean FC across all targets from the mean FC of each individual target. Thus, the fingerprint plots indicate how the basolateral amygdala connects to the targets in each sample, accounting for average differences in connectivity.

## Results

### Anatomically defined OTC

A 2 (sample) x 5 (OTC section) ANOVA was conducted to assess how basolateral amygdala connectivity to the occipitotemporal cortex changes across development. Adults showed significantly more connectivity to the OTC than did neonates, evidenced by a significant main effect of sample, $F(1,390) = 22.42$, $p = 3.08 \times 10^{-6}$. Connectivity to OTC also exhibited topographic differences, evidenced by a main effect of OTC section, $F(4,390) = 20.97$, $p = 1.13 \times 10^{-15}$. More specifically, across samples, connectivity to each of the sections decreased on a gradient from anterior to posterior, with significantly more connectivity to OTC 5 than OTC 4 ($t(79) = -4.44$, $p_{HB} = 1.45 \times 10^{-4}$), to OTC 4 than OTC 3 ($t(79) = -2.55$, $p_{HB} = 0.03$), to OTC 3 than OTC 2 ($t(79) = -4.04$, $p_{HB} = 5.01 \times 10^{-4}$), and to OTC 2 than OTC 1 ($t(79) = -2.06$, $p_{HB} = 0.04$). See S1 Table for all OTC statistical comparisons.

Importantly, the sample x OTC interaction was also significant, $F(4,390) = 13.22$, $p = 4.15 \times 10^{-10}$, revealing topographic connectivity differences in adults but not in neonates. To probe this interaction post hoc, a one-way ANOVA was conducted separately for each sample across the OTC sections (Fig 1B). Whereas the adult sample showed a significant main effect of OTC

section ($F(4,195) = 28.76$, $p = 8.63$ x $10^{-19}$), the neonate sample did not $F(4,195) = 2.05$, $p = 0.09$). Adults showed decreasing connectivity on a gradient from OTC 5 to OTC 4 ($t(39) = -2.91$, $p_{HB} = 0.01$), OTC 4 to OTC 3 ($t(39) = -1.99$, $p_{HB} = 0.05$), OTC 3 to OTC 2 ($t(39) = -6.06$, $p_{HB} = 2.60$ x $10^{-6}$), and OTC 2 to OTC 1 ($t(39) = -3.53$, $p_{HB} = 3.20$ x $10^{-3}$). Although a main effect of OTC section was not observed in neonates, planned t-tests were run to quantify a gradient: neonates showed differentiation between OTC 5 and OTC 4 ($t(39) = -3.33$, $p_{HB} = 0.02$), but connectivity to the rest of the subsequent OTC sections was not significantly different. The differential patterns of connectivity between adults and neonates is additionally represented in an FC fingerprint plot (Fig 1C); the mean-centered connectivity within all five OTC sections significantly differed between adults and neonates. See S2 and S3 Tables for all within- and between-sample OTC statistical comparisons, respectively.

### Functionally defined regions within OTC

**Parcels.** A 2 (sample) x 11 (functional parcel) ANOVA was conducted to assess how basolateral amygdala connectivity to functional parcels within the occipitotemporal cortex changes across development. Again, there was a significant main effect of sample, $F(1,858) = 19.43$, $p = 1.18$ x $10^{-5}$, and parcel, $F(10,858) = 6.39$, $p = 1.57$ x $10^{-9}$.

Additionally, the sample x parcel interaction was also significant, $F(10,858) = 10.43$, $p = 9.21$ x $10^{-17}$, revealing differential connectivity to the parcels in adults but not in neonates. To probe this interaction post hoc, a one-way ANOVA was conducted separately for each sample across the 11 parcels (Fig 2B). Again, whereas the adult sample showed a significant main effect of functional parcel ($F(10,429) = 13.77$, $p = 4.04$ x $10^{-21}$), the neonate sample did not $F(10,429) = 0.78$, $p = 0.65$). Post-hoc t-tests were only conducted on adults (see S4 Table for all comparisons in adults); of particular note, adults showed significantly different connectivity between parcels within the same OTC section, such as A1 and STG within OTC 4 ($t(39) = -5.68$, $p_{HB} = 8.14$ x $10^{-5}$), and between OFA and V1 within OTC 1 ($t(39) = 6.08$, $p_{H B} = 2.35$ x $10^{-5}$), whereas neonates did not show an effect of functional parcel.

**Categories.** To probe whether the connectivity differences across the parcels could better be attributed to overall function rather than anatomical location, a 2 (sample) x 7 (functional category) ANOVA was conducted to assess how basolateral amygdala connectivity to functional categories changes across development. As before, there was a significant main effect of sample, $F(1,546) = 7.19$, $p = 0.01$, and category, $F(6,546) = 9.02$, $p = 2.08$ x $10^{-9}$.

Additionally, the sample x category interaction was also significant, $F(6,546) = 14.97$, $p = 6.96$ x $10^{-16}$, revealing differential connectivity to the functional categories in adults but not in neonates. To probe this interaction post hoc, a one-way ANOVA was conducted separately for each sample across the 7 categories (Fig 3A). Again, whereas the adult sample showed a significant main effect of functional category ($F(6,273) = 20.42$, $p = 9.90$ x $10^{-20}$), the neonate sample did not $F(6,273) = 0.62$, $p = 0.71$). As depicted in Fig 3A, adults showed more connectivity to parcels that functionally process faces, bodies, objects, and high-level auditory processing, and less connectivity to parcels that functionally process scenes and primary auditory and visual cortex. Neonates showed undifferentiated connectivity across categories. As indicated in the FC fingerprint plot (Fig 3B), all seven functional categories exhibited significant between-group differences in mean-centered connectivity patterns. See S5 and S6 Tables for all statistical comparisons within adults and between samples, respectively.

### Discussion

Investigating the functional connectivity between the amygdala and occipitotemporal cortex will help us better understand the amygdala's role in perceiving and processing emotional

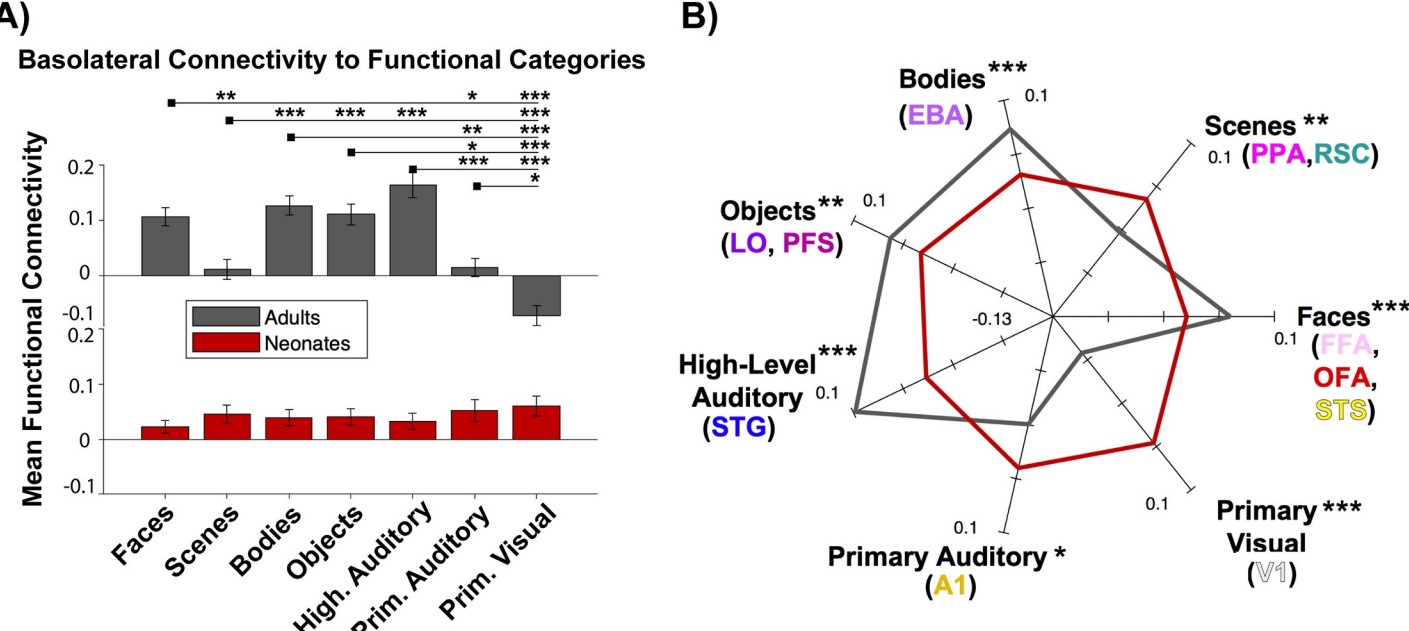

**Fig 3. Functional connectivity to functional categories.** (A) Bar plot of mean functional connectivity to each of the 7 functional categories, with adults in gray and neonates in red. Error bars are standard error of the mean. $*p<0.05$, $**p<0.01$, $***p<0.001$ (C) FC fingerprint plot depicting the pattern of connectivity of both samples. Axes are mean centered FC values for each sample. Parentheses show which of the 11 parcels were included in each category. Asterisks denote significance between groups for each category.

visual stimuli, which has ecological relevance and certainly changes across development. Many functionally specialized visual regions exist within occipitotemporal cortex, but it was previously unknown how connectivity to these regions develops from birth in humans. Previous work in macaques had revealed connections between the lateral and basal amygdala subnuclei and the occipitotemporal cortex, noting a rostrocaudal topographic organization of the connections (e.g., [22]) and refinement across development (e.g., [23]). In this paper, we explored this topographic organization in humans using functional connectivity, and further investigated specific functional cortical areas located within the occipitotemporal region that may contribute to the observed pattern of connectivity.

In our study, connectivity between the basolateral amygdala and occipitotemporal cortex in human adults decreased on a gradient from anterior to posterior, replicating the finding in macaques. However, the connectivity in neonates was largely undifferentiated, suggesting that the topographic organization in adulthood is not yet present at birth. Splitting the cortex into functionally defined parcels allowed us to further hone in on the developmental changes in this pattern. If the gradient of connectivity was reliant on anatomical location (e.g., cortex closer to the amygdala is more functionally connected), then splitting the cortex into parcels should have revealed a comparable gradient. Instead, the parcels had varied connectivity with the amygdala in adults, even when anatomically located in the same OTC section. For instance, within more anterior regions of the OTC, connectivity was driven more by connections with STG (known for processing high-level auditory information, e.g., speech) than with adjacent A1 (primary auditory cortex). Similarly, in posterior OTC, lower connectivity in adults was driven more by connections with V1 (primary visual cortex) than by connections with OFA (known for processing faces). This would suggest that functional processing of the cortex contributes to the development of connectivity between the amygdala and OTC: adults showed

more connectivity to high-level sensory regions (i.e., regions processing faces, bodies, objects, high-level audition) relative to primary sensory regions (i.e., V1, A1). Conversely, neonates had similar connectivity to all functional parcels and categories, with not much differentiation among them. Interestingly, V1 showed the largest developmental difference between the two samples, with positive connectivity in neonates and negative connectivity in adults. These results are in line with studies in macaques (e.g., [23, 24]) where both adult and infant macaques showed comparably high amygdalar connectivity with anterior temporal cortex, but only infants showed additional connections to posterior OTC regions. Human neonates and adults in the present study also showed relatively high connectivity to anterior OTC, but only neonates showed high connectivity with posterior OTC.

The present study also revealed noticeable differences in amygdalar connectivity to each of the distinct functional categories in adults, where basolateral connectivity was highest with face, body, object, and high-level auditory regions. These results directly align with connectivity differences between adults and neonates observed to each of the OTC sections and parcels. For instance, adults showed higher connectivity to each of the OTC sections than did neonates, and the parcels within those sections driving the increased connectivity (i.e., STG in OTC 4; STS and PFS in OTC 3; EBA, LO, and FFA in OTC 2, and OFA in OTC 1) are all associated with either face-, body-, object-, or high-level auditory processing. This pattern of differential connection strength was largely absent in neonates, who showed similar strength of connection to almost all of the functional regions. Unlike the adults, who showed higher amygdalar connectivity with high-level visual and auditory regions than did neonates, neonates showed higher connectivity with primary visual and auditory cortex than did adults. This would suggest that neonates parse the emotional content of their surroundings at a very basic level of processing, and develop emotional associations with high-level functional categories only after maturation.

In contrast, previous work in infants has found similarities in basolateral amygdalar functional circuitry in 3-month-old infants as in adults [37]; although informative, the present study examines neonates with gestational age between 37–44 weeks and thus offers even earlier insight. Our results corroborate a recent study on resting-state connectivity of the whole amygdala, in which the neonatal sample showed positive FC (not adult-like) to primary auditory cortex but adult-like positive FC with the nearby parahippocampus [56]. Likewise, neonates in our study had higher connectivity to primary auditory cortex than adults, and PPA was one of the few regions showing similar connectivity between the two groups.

Given that the neonates showed largely undifferentiated connectivity of the basolateral amygdala with various functional regions compared to adults, but functional organization appears more adult-like after a few months in other studies, we posit that adult-like connectivity between the basolateral amygdala and functional regions of the OTC is not present at birth and instead requires at least some experience (i.e., a few months) to develop. For instance, a study on the development of high-level visual cortex showed that 4-6-month-old infants had similar spatial organization of functional categories (i.e., faces, scenes) as adults, but immature selectivity within those regions [57]. Further exploration into the development of selectivity in the occipitotemporal cortex found that increased performance in behavioral tasks of face-, object-, and symbol-naming in 4-6-year-olds was accompanied by decreased activation of associated cortical regions to their nonpreferred categories [58]. Coupled with these findings, the present results align with the hypothesis that refinement and pruning of connections typically occur after relevant experience (i.e., cognitive milestones) and neural specialization occur [59–61]. In fact, studies of cortical maturation have noted the temporal lobe to be one of the last regions to mature [59], with the ventral visual stream maturing in hierarchical steps organized by increasing perceptual complexity until the end of childhood [62, 63]. Experience with

the environment postnatally may lead to notable age-related changes, and activity-dependent interactions between cortical regions may fine-tune their functionality. For instance, experience with visual words and increased memory performance for visual categories (e.g., faces, scenes, objects) have been shown to be correlated with larger activation of their respective cortical areas (e.g., [64, 65]); single-unit cell recordings in monkeys have suggested that this type of categorical training or experience leads to a greater proportion of responsive cells in inferotemporal cortex compared to controls [66]. The functional maturation of the occipitotemporal regions that were studied here may contribute to 1) the continued refinement of the amygdalar subregions (indeed the size of several amygdalar nuclei change from infancy to adulthood [67], 2) the amygdalar connections to these regions, and 3) the functional specialization of occipitotemporal regions themselves (e.g., in line with Interactive Specialization theories of development; [68]).

One notable limitation of the present study is that we use functional parcels originally defined in adults, and recognize that overlaying them onto neonates may have the potential to overestimate or mischaracterize certain cortex. However, we used ANTs to register the functional parcels to each neonate's native space, which has been shown to be highly effective and reliable [69]. These limitations can only be overcome by functionally defining regions of interest in each individual, which may not be reliable in a sample of newborns (see [70] for a review of neuroimaging methods in adults vs. neonates), or perhaps with longitudinal studies that can functionally localize regions using task-based fMRI at a later age and register them to the same individual's connectivity scan at an earlier age (e.g., [71]). Another avenue of future research is in exploring how the maturation of amygdala-occipitotemporal connectivity is affected by connectivity with other brain regions; for instance, the prefrontal cortex plays a large role in emotion regulation and also has strong connections and resting-state coupling with amygdalar nuclei (e.g., [72]). While we focused on occipitotemporal connections here, exploring related connections in the frontal cortex and would be an interesting question for future research.

Overall, the present experiments make apparent a decreasing pattern of connectivity between the amygdala and posterior aspects of occipitotemporal cortex, evidence for which has been repeatedly shown in macaques but was otherwise lacking in humans. Further, we contrast adult data with a sample of neonates scanned within one week of birth, to gauge what connectivity exists primitively, prior to extensive experience with the world. Additionally, we identify putative functional areas in the ventral visual stream that might be driving the observed pattern of connectivity changes. This work has important clinical applications: Given the role of the amygdala in many psychiatric disorders–many of which have early onsets, such as autism and anxiety (e.g., [35, 73–76])–it is crucial to fully understand how the amygdala connects to the rest of the brain across early development. The developmental progression of connectivity between the amygdala and occipitotemporal cortex in typically-developing humans can help us better understand developmental disorders or deficits implicated when these connections are abnormal or lacking. Further research can seek to explore this connectivity in patient populations, classify differences between patients and controls, and offer new diagnostic or treatment interventions.

## Supporting information

**S1 Fig. BaLa overlap with dHCP DrawEm amygdala label in a representative neonate.** (left) Coronal and axial slices depicting the basolateral amygdala (yellow) as defined by automated segmentation (Saygin et al., 2017), overlaid on the whole amygdala as defined by dHCP's DrawEm label (red). Overlap shown in orange (proportion overlap across all neonates:

0.76 ± 0.11)
(TIF)

**S2 Fig. OTC label comparison in a representative individuals.** (left) neonate, (right) adult. Dark blue = OTC 5 (anterior), blue = OTC 4, light blue = OTC 3, lime green = OTC 2, dark green = OTC 1 (posterior).
(TIF)

**S1 Table. OTC connectivity differences collapsed across samples.** t-test results and corresponding p-values comparing mean connectivity between each OTC section, collapsed across adults and neonates.
(DOCX)

**S2 Table. OTC connectivity differences within each sample.** t-test results and corresponding p-values comparing mean connectivity between each OTC section, separately for adults and neonates.
(DOCX)

**S3 Table. OTC connectivity differences between samples.** t-test results and corresponding p-values comparing mean-centered connectivity between adults vs. neonates in each OTC section, from 5 (anterior) to 1 (posterior). See Fig 1C in main manuscript.
(DOCX)

**S4 Table. Connectivity differences between functional parcels in adults.** t-test results and corresponding p-values comparing mean connectivity between each functional parcel in adults.
(DOCX)

**S5 Table. Connectivity differences between functional categories in adults.** t-test results and corresponding p-values comparing mean connectivity between each functional category in adults.
(DOCX)

**S6 Table. Functional category connectivity differences between samples.** t-test results and corresponding p-values comparing mean-centered connectivity between adults vs. neonates for each functional category. See Fig 3B in main manuscript.
(DOCX)

**S1 File. List of anatomical labels combined to create OTC.**
(PDF)

## Acknowledgments

Analyses were completed using the Ohio Supercomputer Cluster (https://www.osc.edu). We would like to thank the Human Connectome Project (https://www.humanconnectome.org) and developing Human Connectome Project (http://www.developingconnectome.org), David Osher for comments and suggestions, and members of Z-Lab (Saygin Developmental Cognitive Neuroscience Lab) for feedback and comments.

## Author Contributions

**Conceptualization:** Heather A. Hansen, Zeynep M. Saygin.

**Data curation:** Jin Li.

**Formal analysis:** Heather A. Hansen, Jin Li, Zeynep M. Saygin.

**Funding acquisition:** Zeynep M. Saygin.

**Investigation:** Heather A. Hansen, Zeynep M. Saygin.

**Methodology:** Heather A. Hansen, Jin Li, Zeynep M. Saygin.

**Supervision:** Zeynep M. Saygin.

**Visualization:** Heather A. Hansen, Zeynep M. Saygin.

**Writing – original draft:** Heather A. Hansen.

**Writing – review & editing:** Jin Li, Zeynep M. Saygin.

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
