## [Decision Letter · Decision Letter 0]

24 Sep 2020

PONE-D-20-22317

Adults vs. neonates: Differentiation of functional connectivity between the basolateral amygdala and occipitotemporal cortex

PLOS ONE

Dear Dr. Hansen,

Thank you for submitting your manuscript to PLOS ONE. After careful consideration, we feel that it has merit but does not fully meet PLOS ONE’s publication criteria as it currently stands. Therefore, we invite you to submit a revised version of the manuscript that addresses the points raised during the review process.

We look forward to receiving your revised manuscript.

Kind regards,

Paul Hinckley Delano, Ph.D.

Academic Editor

PLOS ONE

Journal Requirements:

Reviewers' comments:

Reviewer's Responses to Questions

**Comments to the Author**

1. Is the manuscript technically sound, and do the data support the conclusions?

Reviewer #1: Yes

Reviewer #2: Yes

2. Has the statistical analysis been performed appropriately and rigorously? 

Reviewer #1: Yes

Reviewer #2: Yes

3. Have the authors made all data underlying the findings in their manuscript fully available?

Reviewer #1: Yes

Reviewer #2: Yes

4. Is the manuscript presented in an intelligible fashion and written in standard English?

Reviewer #1: Yes

Reviewer #2: Yes

5. Review Comments to the Author

Reviewer #1: The paper by Hansen et al. analyzed the functional connectivity of the basolateral region of the amygdala to each of five regions along the rostro-caudal axis of the occipitotemporal cortex in adults and in neonates, using data from the HCP and dHCP databases. The study focuses on the amygdalar nucleus that was found to undergo the most age-related changes in a previous study from the same lab, which is also the nucleus with strong connections to the visual system.

They document a decreasing rostro-caudal gradient of connectivity to the anatomically defined regions, to functional areas along that axis, and to functional categories that could involve structures and areas located in more than one of the anatomical regions. None of these gradients was present in neonates, with the exception to some degree of the anatomical gradient. These results are comparable to a similar gradient found in macaque tracer studies.

The results are clearly presented, and effort was made to make the two sets of data and the anatomical regions in adults and neonates comparable. I acknowledge that my command of the statistics involved in the processing of imaging data is limited, so I am not going to delve into that aspect. But the differences between the two sets of data are statistically robust. There are some intriguing differences -not addressed in the text- in the degree of functional connectivity from the amygdala to some of the regions; for example, to regions 4 & 3. Are they related to any particular area?

The manuscript is clearly written; however, I think the Results section would gain in clarity if a summary sentence was added before the detailed statistics of each set of data.

Minor points:

1. Is the adult template used to define the amygdala subnuclei applicable to neonates?. There is a study that shows differences in size and in the relative area of several amygdalar nuclei in infants as compared to adults (Proc Natl Acad Sci U S A. 2018 Apr 3; 115(14): 3710–3715.

2. The ventral visual stream is highly immature at birth. Some reference could be made to the maturation timetable of some of the regions depicted in the study.

3. When discussing some other work in infants, reference could be made of Biol Psychiatry Cogn Neurosci Neuroimaging. 2019 Jan; 4(1): 62–71., that also studied the amygdala functional connectivity during infancy, including neonates.

4. In figure 3, A1 seems to be misplaced: It looks as if it is on the frontal lobe, or near the central sulcus.

Reviewer #2: General comments:

In this study was investigated the changes in functional connectivity between the basolateral amygdala and the occipitotemporal cortex that are induced in human brain maturation between neonates and adult subjects. The results suggest that the functional connectivity between these two brain structures is not yet differentiated in neonates.

The results are interesting and will be an important contribution to understand how the emotional circuit matures in the human brain. The authors used a correct experimental design and the manuscript is well written.

Minor concerns:

The following points should be addressed in the discussion section:

1) The functional connectivity between the amygdala and the neocortex somehow tells us how the human brain emotionally translates the world that it lives. So, the authors could suggest how the changes in functional connectivity they found could affect the emotional processing of sensory stimuli that are perceived from the environment. How does the brain of neonates emotionally translate the world? what is lost or possibly gained when the brain matures into adulthood? What your results tell you about these questions.

2) The frontal cortex has a fundamental role in the amygdala activity, I understand that the authors did not explore this part of the story in this investigation. However, this is a key piece of the puzzle that should not be excluded when interpreting the results. GABAergic activity is immature in the frontal cortex of neonates, how can this affect the maturation of functional connectivity between basolateral amygdala and occipitotemporal cortex?

3) How can the environment affect neural circuits and induce the functional connectivity changes found in this study?

4) This study was conducted only in women. So, how sex hormones could contribute to the changes in functional connectivity found in this study.

6. PLOS authors have the option to publish the peer review history of their article (what does this mean?). If published, this will include your full peer review and any attached files.

Reviewer #1: No

Reviewer #2: **Yes: **Alexies Dagnino-Subiabre

---

## [Author Response · Author response to Decision Letter 0]

29 Sep 2020

Point-by-point responses to each of the reviewer and editor comments can be found in the "Response to Reviewers" file.

---

## [Editor Report · Decision Letter 1]

6 Oct 2020

Adults vs. neonates: Differentiation of functional connectivity between the basolateral amygdala and occipitotemporal cortex

PONE-D-20-22317R1

Dear Dr. Hansen,

We’re pleased to inform you that your manuscript has been judged scientifically suitable for publication and will be formally accepted for publication once it meets all outstanding technical requirements.

Kind regards,

Paul Hinckley Delano, Ph.D.

Academic Editor

PLOS ONE
---

## [Editor Report · Acceptance letter]

9 Oct 2020

PONE-D-20-22317R1 

Adults vs. neonates: Differentiation of functional connectivity between the basolateral amygdala and occipitotemporal cortex 

Dear Dr. Hansen:

I'm pleased to inform you that your manuscript has been deemed suitable for publication in PLOS ONE. Congratulations! Your manuscript is now with our production department. 

Kind regards, 

on behalf of

Dr. Paul Hinckley Delano 

Academic Editor

PLOS ONE